# Relative Weights of Physical Strength Factors in Sports Events: Focused on Similarity Sports Events Group According to the Sports Physiological View

**Kyoung-Hyun Lee [1], Jin-Seok Lee [2], Byung-Chan Lee [3] and Eun-Hyung Cho [2],***

[1] Center for Sport Science in Gwangju, 278, Geumhwa-ro, Seo-gu, Gwangju 62048, Korea; brion118@skku.edu
[2] Department of Sports Science, Korea Institute of Sport Science, 727, Hwarang-ro, Nowon-gu, Seoul 01794, Korea; js0420@kspo.or.kr
[3] Department of Physical Education, Sichuan Agricultural University, Ya'an 625014, China; wongkei@korea.ac.kr
* Correspondence: ehcho@kspo.or.kr; Tel.: +82-970-9564

**Abstract:** The purpose of this study was to investigate the relative weights of physical strength factors in sports events. We selected 16,645 people as a sample group who participated in physical fitness measurements through eight sports science centers across the country from 2016 until August of 2018, and divided into four sports types depending on the sports physiological view: type A: short-term muscular power and short-term muscular endurance, type B: mid-term muscular power, type C: long-term cardiorespiratory endurance, type D: coordination capability (CC), agility, flexibility, and balance. Categorized the performance level into excellent athletes and non-excellent athletes, and standardized (T-score) the measured value after considering sex, age and sports type group. Used logistic regression analysis for the method of analysis, and calculated the relative weights of physical strength factor with different sports by using Wald value which was calculated from logistic regression analysis. As a result, the relative weights of physical factor in type A were power 30%, muscular power (MP) 18%, CC 16%, agility 11%, flexibility 10%, cardiorespiratory endurance (CE) 1%, and balance 0%. The relative weights of physical factor in type B were muscular endurance (ME) 43%, MP 25%, power 20%, balance 9%, CE 2%, flexibility 1%, agility 0%, and CC 0%. The relative weights of physical factor in type C were ME 41%, CE 37%, power 10%, agility 8%, flexibility 2%, CC 2%, ME 0%, and balance 0%. Need more specific classification standard for type D sports. Hope the results of this study were used to measure physical fitness level and used as baseline data for recruiting future talents.

**Keywords:** sports events; physiological view; physical strength factor; relative weight

## 1. Introduction

It's a proven fact through many studies that physical strength is a vital factor to improve performance in sports. However, the importance of physical strength differs by sport event. It is necessary to find out the physical factors that affect the performance of each sport.

If one examines the studies about finding out the physical factors that are related with the performance, you can see that they are sorted by two big categories which are theoretical studies about the literature, or content validity with collections of expert opinions [1] and studies based on measurement of physical fitness data [2,3].

The first category has strength in internal validity and contents as it applies literature and the opinions of experts, while the other category has strength in objectivity as it applies physical fitness data. However, both methods can verify the priorities of important physical factors of each sport,

and has limitations in finding out the magnitude of the importance. For example, for a judo athlete one can verify that power is a more important factor than muscular endurance, but we cannot verify how more important it is. In other words, it has weakness in assigning relative weights to each factor.

Verifying relative weights of physical factors can quantify the physical factors that individuals have and can be used as important baseline data when recruiting future talent [4]. If you look at the selection process of future talent within countries, they apply different importances of performance, physique, and physical factors to each sport, and also apply different importances of physical factors to each sport according to detailed measurement items [5]. However, even though verifying relative weights of physical factors is a very crucial data which can be used for recruiting, precedent research is not enough and inadequate. One of the reasons why verifying relative weights of physical factors in each sport is not enough and inadequate, is the problem of the amount of data. To secure the validity of the verified weights, we need data analysis based on big data and collecting enough physical data of elite athletes is just too difficult unless it happens to be available at a national level.

Meanwhile, the Korean Ministry of Culture is operating a pilot project to provide sports science support to local elite athletes which is provided by the Korea Institute of Sports Science which is applying it to national team athletes to improve the performance of the local athletes since 2015. Year 2018 as it is now, there are eight local sports science centers operating across the country and their goal is to provide sports science support to 1400 athletes at each center annually. Therefore, if we use accumulated measurement data from the local sports centers, we can estimate the validity of verified relative weights of physical factors in each sport. Thus, the purpose of this study is to verify the relative weights of physical factors in each sport using accumulated big data measured at the local sports science centers.

## 2. Subjects and Statistical Analysis

### 2.1. Data Collection

To achieve the purpose of this study, selected 16,645 athletes as a population who participated in physical fitness measurement from eight sports science centers across the country from 2016 until August of 2018. Table 1 shows the sex and number of the populations by age groups.

**Table 1.** Sex and number of populations by age groups.

| Sex | Elementary School (Age: 8–13) | Middle School (Age: 14–16) | High School (Age: 17–19) | Adults (Age: 20–30) | Total |
|---|---|---|---|---|---|
| Male | 1627 | 3721 | 3289 | 2430 | 11,067 |
| Female | 897 | 1519 | 1705 | 1457 | 5578 |
| Overall | 2524 | 5240 | 4994 | 3887 | 16,645 |

### 2.2. Classification Criteria of Similar Sports Types

Based on the 'Research task report on local sports science 2017' and the development of physical training in each sport [6], they are categorized into five types depending on the sports physiological viewpoint. Concretely, the five types are sports events focused on short-term muscular power and short-term muscular endurance (Type A, 24 sports), mid-term muscular endurance (Type B, 19 sports), long-term cardiorespiratory endurance (Type C, 10 sports), agility, coordination capability, balance (Type D, 25 sports), and speed endurance (Type E, 11 sports). On the basis of this, in this research, through the discussion of experts, we excluded sports events focused on speed endurance (Type E) to fit the character of research materials and recategorized as four events (A, B, C, D). Table 2 shows the classification standard of similar sports events depending on the sports physiological viewpoint.

**Table 2.** Classification criteria of similar sports type grouped according to the sports physiological viewpoint.

| Type | Classification Criteria | Detailed Sports Type |
|---|---|---|
| A | Physical factors that decide performance: short-term muscular power, short-term muscular endurance Short-term muscular power and anaerobic endurance training sports Anaerobic 40~100%, aerobic 0~60% sports Super Short time (less than 10 s), shot-term performance (10~180 s) sports | Track & field (100~800 m), athletics (jumping), athletics (throwing), cycle (200~1000 m), short-track (500~1500 m), diving, gymnastics, judo, wrestling, ssireum, golf, fencing, kendo, weightlifting, bodybuilding, sports climbing, snowboarding (cross), alpine skiing, freestyle skiing (cross) |
| B | Physical factors that decide performance: mid-term muscular endurance Mid-term muscular endurance and aerobic training sports Anaerobic 10~40%, aerobic 60~90% sports Short time (3~20 min), mid time performance (21~60 min) sports | Track & field (1000~10,000 m), cycling (4000 m), short-track (3000 m), speed skating, boxing, taekwondo, synchronized swimming, canoe, water polo, kayak, rowing, kickboxing, wushu, aerobics, race walking, inline skating, swimming, fin swimming |
| C | Physical factors that decide performance: long-term cardiorespiratory endurance Long-term muscular endurance and long-term cardiorespiratory endurance training sports anaerobic 0~10%, aerobic 90~100% sports Long-time (1~4 h) performance sports | Modern pentathlon, marathon, cycling (MTB), cycling (road), yachting, biathlon, triathlon, cross country skiing, Nordic skiing, ice hockey, cycling (long distance) |
| D | Physical factors that decide performance: agility, coordination capabilities, flexibility and balance Agility, coordination capabilities training sports ATP-PCr 60~90%, glycolysis 0~20% sports | Basketball, baseball, volleyball, table tennis, handball, tennis, badminton, cricket, shooting, archery, curling, cycling (BMX), field hockey, figure skating, snowboarding (half pipe), rugby, soccer, sepaktakraw, softball, archery, squash, tennis, billiards |

A: short-term muscular power and short-term muscular endurance, B: mid-term muscular endurance. C: long-term cardiorespiratory endurance, D: agility, coordination capability, balance. Reference: Research task report on local sports science 2017 from the development of physical training in each sports event [6].

## 2.3. Assessment Tools to Physical Fitness

Physical factors chosen to achieve the purpose of this study are strength, muscle endurance, power, cardiorespiratory function, agility, flexibility, balance and coordination, and each method of measurement is shown in Table 3.

## 2.4. Categorized the Performance Level

The performance level is categorized into excellent athletes and non-excellent athletes. Categorization criteria were first selected as domestic and international standing status and second, they were classified according to close support status. Close support is a program that provides sports science support from the local sports centers to those with high performance and with high chance of winning the medals. Table 4 shows the sports events and sample sizes in athletes.

## 2.5. Data Standardized and Sampling

The age of the sample group for this research is from elementary through high school students and adults, sex is both. Therefore, the use of raw data means age and sex could be variables. For example, an excellent elementary athlete could have better physical ability than a non-excellent adult athlete or an excellent female athlete could have better physical ability than a non-excellent male athlete. To solve

this problem, I considered sex, age, sport events, and calculated the personalized T-score which is standardized to each group. Calculation formula of T-score is as follows.

$$T = \left(\frac{X - M_i}{S_i}\right) \times 10 + 50 \tag{1}$$

In Equation (1), $X$ means individuals' original score, and $M_i$ means the average of sex, age, and sport type group. $S_i$ means the standard deviation between sex, age and sport type group. After calculating the T-score of individuals, if it scores over than 90 and less than 10, I regarded them as churn values and deleted them. For the physical score, used average of each measurement items to calculate.

**Table 3.** Assessment tools according to physical fitness.

| Sex | Assessment Tools |
|---|---|
| Strength | Grip Strength (kg)<br>Back Strength (kg) |
| Muscle endurance | Sit-Up (count/60 s)<br>Push-Up (count/60 s) |
| Power | Sargent Jump (cm)<br>Standing Long Jump (cm) |
| Cardiorespiratory function | FEV$_1$ (%)<br>20-m Shuttle Run Test (count) |
| Agility | Reaction Time (1/1000 s)<br>Side-Step Test (count/20 s) |
| Flexibility | Trunk Flexion (cm)<br>Trunk Extension (cm) |
| Balance | One Leg Balance with Eyes Closed (s)<br>Dynamic Balance (s/min.) |
| Coordination | Eye-Hand Coordination (s) |

FEV: forced expiratory volume.

**Table 4.** Sports events and sample sizes in athletes.

| Sex | A | B | C | D | Overall |
|---|---|---|---|---|---|
| Excellent | 530 | 378 | 77 | 888 | 1873 |
| Non-excellent | 3421 | 3138 | 642 | 7571 | 14,772 |
| Overall | 3951 | 3516 | 719 | 8459 | 16,645 |

A: short-term muscular power and short-term muscular endurance, B: mid-term muscular endurance. C: long-term cardiorespiratory endurance, D: agility, coordination capability, balance.

In Table 4, as suggested earlier, the number of sports events group excellent athletes and non-excellent ones differs. Among a total number of 16,645 athletes, excellent athletes are 1873, which represents 11.3%. When a logistic regression model is applied, if there is a big difference in the sample sizes of a dependent variables, so the fidelity of the model could be low. Concretely, the non-excellent athletes sample size is larger, so there is a possibility of the model that might focus on classification prediction of the non-excellent athletes. While this makes more classification of non-excellent athletes more accurate, it lowers however the accuracy of the classification of excellent athletes. To solve this problem, we implemented stratified randomization random sampling which considers sex and age [7]. Table 5 shows the characteristics of the finally selected sample groups.

**Table 5.** Sports events and sample sizes in athletes.

| Sex | A | B | C | D | Total |
|---|---|---|---|---|---|
| Excellent | 231 | 156 | 46 | 304 | 737 |
| Non-excellent | 231 | 156 | 46 | 304 | 737 |
| Total | 462 | 312 | 92 | 608 | 1474 |

A: short-term muscular power and short-term muscular endurance, B: mid-term muscular endurance. C: long-term cardiorespiratory endurance, D: agility, coordination capability, balance.

## 2.6. Statistical Analysis

We used Excel 2014 (Microsoft, Redmond, WA, USA) to organize the data and calculate T-scores. We implemented logistic regression analysis to explore the physical factors of performance determinants. For calculation of relative weigh values, we used logistic regression analysis Wald values so the sum of each physical factor become 100% through the proportions. We used SPSS version 25.0 (SPSS Inc., Chicago, IL, USA) for statistical processing, and set the statistical significance level at 0.05.

## 3. Results

### 3.1. Results of Logistic Regression Goodness of Fit Model

Table 6 presents the results of the goodness of fit of our logistic regression analysis by sport events. The constants of the type A, B and C models appear to have a $X^2$ significance probability value that is statistically reasonable—between 2LL and the theoretical model that the researchers set (intercept model-theory model), on the other hand, type D appear to have no significance probability. Nagelkerke $R^2$ generally appeared low in all sport type groups, but in case of logistic regression analysis, the sums of the coefficient of determinations differ depending on the value of a dependent variable, but still that value is generally low as well. If one looks at the classification precision, it shows that type A is 58.9%, B is 65.4%, C is 69.6%, and D is 55.4%.

**Table 6.** The goodness of fit of the logistic regression analysis by sport events.

| Validation Method | | A | B | C | D |
|---|---|---|---|---|---|
| -2LL | | 616.3 | 383.7 | 109.0 | 828.7 |
| $X^2$ | | 24.1 | 48.9 | 18.5 | 14.1 |
| *df* | | 8 | 8 | 8 | 8 |
| *p* | | 0.002 | 0.001 | 0.018 | 0.079 |
| *Nagelkerke $R^2$* | | 0.068 | 0.193 | 0.243 | 0.031 |
| Classification precision | Excellent | 57.1% | 65.4% | 67.4% | 53.6% |
| | Non-excellent | 60.6% | 65.4% | 71.7% | 57.2% |
| | overall | 58.9% | 65.4% | 69.6% | 55.4% |

A: short-term muscular power and short-term muscular endurance, B: mid-term muscular endurance. C: long-term cardiorespiratory endurance, D: agility, coordination capability, balance.

### 3.2. Results of Relative Weight of Physical Fitness

To calculate the relative weighs of physical factors by sport event group, we used Wald values from the logistic regression analysis. Concretely, these were calculated through a proportional expression so that the sum of Wald values is 100% for each physical factor. The relative weighs resulting from this are shown in Table 7.

For A type, the values are power 30%, strength 18%, coordination 16%, agility 11%, flexibility 10%, cardiorespiratory function 1%, balance 0%. For B type they are muscle endurance 43%, strength 25%, power 20%, balance 9%, cardiorespiratory function 2%, flexibility 1%, agility 0%, coordination 0%. For C type, they appear to be muscle endurance 41%, cardiorespiratory function 37%, power 10%, agility 8%, flexibility 2%, coordination 2%, strength 0%, balance 0%. For D type, they are coordination

29%, flexibility 24%, strength 21%, balance 16%, muscle endurance 7%, power 2%, cardiorespiratory function 1%, agility 0%.

**Table 7.** The relative weights of physical fitness factors.

| Physical Fitness Factor | Sport Type Group | | | |
|---|---|---|---|---|
| | A | B | C | D |
| Strength | 18% | 25% | 0% | 21% |
| Muscle endurance | 14% | 43% | 41% | 7% |
| Power | 30% | 20% | 10% | 2% |
| Cardiorespiratory function | 1% | 2% | 37% | 1% |
| Agility | 11% | 0% | 8% | 0% |
| Flexibility | 10% | 1% | 2% | 24% |
| Balance | 0% | 9% | 0% | 16% |
| Coordination | 16% | 0% | 2% | 29% |

A: short-term muscular power and short-term muscular endurance, B: mid-term muscular endurance, C: long-term cardiorespiratory endurance, D: agility, coordination capability, balance.

### 3.3. Results of Logistic Regression Analysis

As a result, a statistical significance about physical factors of performance by sport event group is obtained. For type A, only the factor of power (Wald = 6.153, $p = 0.013$) appeared to have statistical significance, for type B, the factors of strength (Wald = 6.533, $p = 0.011$), muscular endurance (Wald = 11.298, $p = 0.001$, and power (Wald = 5.215, $p = 0.022$) appeared to have statistical significance. For type C, the factors of muscular endurance (Wald = 4.901, $p = 0.027$) and cardiorespiratory function (Wald = 4.462, $p = 0.035$) appeared to have statistical significance, whereas for type D none of the physical factors appeared to have statistical significance (Table 8).

**Table 8.** Results of physical fitness factors on performance by sport type (logistic regression analysis).

| Type | Variables | B | SE | Wald | df | p | Exp(B) |
|---|---|---|---|---|---|---|---|
| A | Strength | 0.021 | 0.011 | 3.711 | 1 | 0.054 | 1.021 |
| | Muscle endurance | 0.020 | 0.012 | 2.837 | 1 | 0.092 | 1.020 |
| | Power | 0.032 | 0.013 | 6.153 | 1 | 0.013 | 1.032 |
| | Cardiorespiratory function | −0.005 | 0.012 | 0.184 | 1 | 0.668 | 0.995 |
| | Agility | 0.002 | 0.015 | 2.145 | 1 | 0.143 | 1.022 |
| | Flexibility | −0.018 | 0.012 | 2.000 | 1 | 0.157 | 0.982 |
| | Balance | 0.003 | 0.010 | 0.094 | 1 | 0.759 | 1.003 |
| | Coordination | −0.020 | 0.011 | 3.279 | 1 | 0.07 | 0.980 |
| B | Strength | 0.037 | 0.015 | 6.533 | 1 | 0.011 | 1.038 |
| | Muscle endurance | 0.053 | 0.016 | 11.298 | 1 | 0.001 | 1.054 |
| | Power | 0.037 | 0.016 | 5.215 | 1 | 0.022 | 1.037 |
| | Cardiorespiratory function | −0.011 | 0.017 | 0.437 | 1 | 0.508 | 0.989 |
| | Agility | 0.000 | 0.017 | 0.000 | 1 | 0.996 | 1.000 |
| | Flexibility | 0.008 | 0.017 | 0.223 | 1 | 0.637 | 1.008 |
| | Balance | 0.021 | 0.014 | 2.299 | 1 | 0.129 | 1.022 |
| | Coordination | 0.002 | 0.014 | 0.024 | 1 | 0.876 | 1.002 |
| C | Strength | −0.008 | 0.035 | 0.054 | 1 | 0.816 | 0.992 |
| | Muscle endurance | 0.089 | 0.040 | 4.901 | 1 | 0.027 | 1.093 |
| | Power | −0.041 | 0.039 | 1.139 | 1 | 0.286 | 0.959 |
| | Cardiorespiratory function | 0.073 | 0.035 | 4.462 | 1 | 0.035 | 1.076 |
| | Agility | 0.033 | 0.035 | 0.895 | 1 | 0.344 | 1.034 |
| | Flexibility | 0.016 | 0.030 | 0.261 | 1 | 0.609 | 1.016 |
| | Balance | −0.001 | 0.025 | 0.001 | 1 | 0.971 | 0.999 |
| | Coordination | 0.011 | 0.023 | 0.209 | 1 | 0.647 | 1.011 |

**Table 8.** *Cont.*

| Type | Variables | B | SE | Wald | df | *p* | Exp(B) |
|------|-----------|-----|-----|------|-----|-----|--------|
| | Strength | 0.016 | 0.010 | 2.255 | 1 | 0.133 | 1.016 |
| | Muscle endurance | −0.009 | 0.011 | 0.743 | 1 | 0.389 | 0.991 |
| | Power | −0.005 | 0.012 | 0.204 | 1 | 0.651 | 0.995 |
| D | Cardiorespiratory function | −0.003 | 0.010 | 0.071 | 1 | 0.789 | 0.997 |
| | Agility | 0.002 | 0.013 | 0.028 | 1 | 0.866 | 1.002 |
| | Flexibility | 0.017 | 0.011 | 2.531 | 1 | 0.112 | 1.017 |
| | Balance | −0.012 | 0.009 | 1.692 | 1 | 0.193 | 0.988 |
| | Coordination | 0.018 | 0.010 | 3.062 | 1 | 0.08 | 1.018 |

## 4. Discussion

As confirmed in many cases from advanced sports countries, physical factors are essential factors in improving athletic performance [8,9]. Objective and accurate physical examination and evaluation of physical strength related to performance improvement by sport are emphasized. In particular, efforts to explore and apply performance-related physical factors in each sport have been attempted due to the problem of specificity in each event [1–3].

Relative weighs of physical factors in each sport can be assessed quantitatively and this might be used as a useful as baseline data for selecting athletes. Nevertheless, it is not easy to see the effort needed to calculate the relative weights of each physical factor. Therefore, this study aimed to compare the importance of physical factors in each sport and calculate the relative weighs of the various physical factors. In order to achieve the purposes of the study, the sports groups were divided into four types according to the sports physiology perspective: A type: sports that focus on short-term muscular power and short-term muscular endurance, B type: sports that focus on mid-term muscular endurance, C type: sports that focus on long-term cardiorespiratory endurance and D type: sports that focus on agility, coordination capabilities, flexibility and balance.

In the type A case, according to the result of our logistic regression analysis, only the power factor had a statistically significant effect on performance, and the relative weights from the Wald values were high in order power, strength, and coordination. In other words, power has the most relevance in type A sports. This is consistent with the importance of power mentioned in the studies of short-distance runners [10] and weightlifters [11] which are type A sports. Serresse and colleagues said that the ability to exert a sudden burst of force, or power, affects the performance of short-distance track and field athletes [10]. Hakkinen and colleagues explained that strength and power are important factors in weightlifters' performance [11].

The relative weight of coordination was 16 percent, the third highest. In this study, the data of eye-hand coordination tests was used, and eye-hand coordination is the technology that identifies visual information of the eye in the brain which responds to the motion information of the hand [12]. This means how accurate and fast an athlete reacts to visually perceived information, and in another study, it was stated that reaction time and eye-hand coordination are highly relevant [13].

In the type B case, according to the logistic regression analysis results, the factors strength, muscle endurance and power had statistically significant effects on performance, and from the the calculation of relative weights using the corresponding Wald values, the results were high in the order of factors strength, muscle endurance and power. In a related study about Type B sports, Shaharudin and Agrawal mentioned that the ability of muscular function, average power and highest power, including muscular and muscular endurance are the main factors affecting the performance of rowers, which is believed to support the results of this study [14]. Also, Hernandez and colleagues reported power, strength, and balance as the determinant factors [15]. The results of this study show that calculated balance factors had relatively small weights compared to the factors of strength, muscle endurance and power, and they represented the fourth highest calculated point at 9%.

In the type C case, according to the logistic regression analysis results, muscle endurance and cardiorespiratory function factors had statistically significant effects, and from the calculation of relative weights using the corresponding Wald values, the results were in the order of muscle endurance, cardiorespiratory function and power. If one looks at the type C events, most of them are long-distance sports, and the importance of endurance performance such as muscular endurance and cardiorespiratory endurance cannot be denied. On the other hand, if we look at the reason for the power factor weight, which is the third highest, Riechman and colleagues have reported that strength and power of the lower body are important factors for endurance performance, and in particular, it is reported that there is a correlation between aerobic capacity and muscular power [16]. In addition, in a study of biathlon athletes, it was reported that maximum oxygen intake has a high correlation with standing long jump and standing high jump performance, and considering that characteristics of biathlon running on inclined planes, anaerobic capacity is described as an important performance factor as well as aerobic capacity [17], so it is believed that power can also be an important factor.

In the type D case, according to the logistic regression analysis results, none of the physical factors had a statistically significant effect, and there was also no statistically significant effect in the Wald values from the analysis results of fidelity of the logistic regression model. It would seem that physical factors fail to explain performance and it is inappropriate to apply information about the relative weights that are calculated on this basis. It can be inferred that this result was due to the broad classification criteria of the type D sports. In this study, similar sports types were grouped from a physiological standpoint and the criteria are classified in detail according to the degree of muscle endurance, power, cardiorespiratory function in the type A, B and C, but in the case of type D, the criteria for classification are agility, coordination, flexibility, balance etc., which are somewhat unclear. Therefore, for type D, it is deemed necessary to classify the sport type by applying more detailed classification criteria, and we look forward to future studies supplementing our resulyts.

This study was carried out to calculate the relative weights of sport event physical factors which are sorted from a physiological standpoint. Our research team is also aware that the classification of types according to the physiological standpoint in this study will be controversial for some sports experts and because of the special nature of each sport, clustering might seem meaningless. However, it is impossible to subdivide every sport using a consistent standard and apply it to the field. This is because for team sports, it needs to be subdivided for each position to apply, and for weight division of sports, it needs to be divided into each weight. Furthermore, one needs to calculate the physical importance of each individual according to performance management style, and apply the training program that suits that individual. Athletes at a national level or above deserve individualized assessment and training programs, but for when recruiting future talent, the objective evaluation process based on generalized evaluation methods should be prioritized. Therefore, it is believed that the clustering of similar sports type groups is necessary at the national level.

The past studies on clustering of similar sports type groups reported relations between physical factors categorized into combat, team, individual, target, challenge, etc. This is a classification that takes into account those with characteristics similar to the content area presented [18]. In the national-level curriculum this study will therefore have significant implications in that it newly classifies sport type groups from a physiological standpoint and calculates the importance of performance and physical factors. More subdivisions are needed for type D, and we look forward to this in future studies.

The study also used statistical techniques to calculate the degree of importance of the physical factors of each sport. The results produced by statistical techniques are objective but will not be an absolute criterion. In order to calculate more relevant weighing factors for each sport, the content validity, including the Delphi method, which reflects expert opinion, should also be reflected in the results.

## 5. Conclusions

The conclusions of this study are as follows: First of all, the relative weighs of type A sports are power 30%, strength18%, coordination 16%, agility, 11%, flexibility 10%, cardiorespiratory function 1%, balance 0%. Secondly, the relative weighs of type B sports are muscle endurance 43%, strength 25%, power 20%, balance 9%, cardiorespiratory function 2%, flexibility 1%, agility 0%, coordination 0%. Thirdly, the relative weighs of type C sports are muscle endurance 41%, cardiorespiratory function 37%, power 10%, agility 8%, flexibility 2%, coordination 2%, strength 0%, balance 0%. D sports should be classified by applying more detailed classification criteria.

**Author Contributions:** K.-H.L. designed and performed the experiments. J.-S.L. and B.-C.L. analysed the data and advanced research, and wrote the manuscript. E.-H.C. contributed to the design and implementation of the research, to perform the experiments and to the analysis of results and to the writing of the manuscript. All authors have read and agreed to the published version of the manuscript.

**Funding:** This study was funded by the Korean Institute of Sport Science.

**Acknowledgments:** The authors would like to express heartfelt thanks to the Korean athletes and coaches who participated in the study.

**Conflicts of Interest:** The authors declare no conflict of interest. Furthermore, the funders had no role in the design of the study; in the collection, analyses, or interpretation of data; in the writing of the manuscript, or in the decision to publish the results.

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
