# Peer review of "Relative Weights of Physical Strength Factors in Sports Events: Focused on Similarity Sports Events Group According to the Sports Physiological View"

_applsci, doi:10.3390/app10249131_

Round 1
Reviewer 1 Report
Consult attachment

Author Response
First of all, thank you for your careful consideration.
*attaced file

Reviewer 2 Report
- Line 38: They should cite bibliography in this paragraph to justify this claim.
- In table 1 they must indicate the age range in each of the categories (elementary school, middle school, etc.) because this age range may be different for other countries.
- They should change "," to ".".
- In the notes of table 2 the symbol "*" appears, however it does not appear in the table: * Reference: Research task report on local sports science 2017 From the development of physical training in each sports event. The word “from” must be lowercase.
- Review the line spaces between the tables and the next paragraph.
- Line 97: It is not appropriate to speak in the first person "I considered".
- Line 105: correct this sentence "If you look at the Table 4".
- They should improve the material and methodology section.
- They should improve the material and methodology section. It should include sections such as subjects, statistical analysis, etc.
- This sentence should go in the statistical analysis: “To calculate the relative weighs of physical factors by sport event group, used Wald value from logistic regression analysis. Concretely, calculated through proportional expression so that the sum of Wald value becomes 100% for each physical factors”.
- The discussion section should begin with the objective of the study.
- No previous studies have been cited that have used the same tests to assess physical condition.
- The results and discussion sections should improve significantly.
Author Response
First of all, thank you for your careful consideration.
* attached file

Round 2
Reviewer 2 Report
Table 2: In the lower part of the table, the type of physiological sports should be described.
Table 3: modify the alignment of the text in the second column.
Table 4, 5, 6 and 7: groups A-D must be defined at the end of the table.
Page 7, table 7: the table numbering is wrong.
Page 7, table 7: describe the variables and type of sports at the end of the table.
It is recommended that the article be analyzed by a professional translator.
Author Response
Thank you for the careful review.
The review contents requested by the reviewer have been completed as much as possible.
Thank you again.
